# Detection of Triacetone Triperoxide (TATP) Precursors with an Array of Sensors Based on MoS_2_/RGO Composites

**DOI:** 10.3390/s19061281

**Published:** 2019-03-13

**Authors:** Qihua Sun, Zhaofeng Wu, Haiming Duan, Dianzeng Jia

**Affiliations:** 1School of Physics Science and Technology, Xinjiang University, Urumqi 830046, China; SUNQIHUA520@163.com; 2Key Laboratory of Energy Materials Chemistry, Ministry of Education, Key Laboratory of Advanced Functional Materials, Xinjiang University, Urumqi 830046, China

**Keywords:** (TATP) precursors, MoS_2_/RGO composites, sensor arrays, synergistic effect

## Abstract

Triacetone triperoxide (TATP) is a self-made explosive synthesized from the commonly used chemical acetone (C_3_H_6_O) and hydrogen peroxide (H_2_O_2_). As C_3_H_6_O and H_2_O_2_ are the precursors of TATP, their detection is very important due to the high risk of the presence of TATP. In order to detect the precursors of TATP effectively, hierarchical molybdenum disulfide/reduced graphene oxide (MoS_2_/RGO) composites were synthesized by a hydrothermal method, using two-dimensional reduced graphene oxide (RGO) as template. The effects of the ratio of RGO to raw materials for the synthesis of MoS_2_ on the morphology, structure, and gas sensing properties of the MoS_2_/RGO composites were studied. It was found that after optimization, the response to 50 ppm of H_2_O_2_ vapor was increased from 29.0% to 373.1%, achieving an increase of about 12 times. Meanwhile, all three sensors based on MoS_2_/RGO composites exhibited excellent anti-interference performance to ozone with strong oxidation. Furthermore, three sensors based on MoS_2_/RGO composites were fabricated into a simple sensor array, realizing discriminative detection of three target analytes in 14.5 s at room temperature. This work shows that the synergistic effect between two-dimensional RGO and MoS_2_ provides new possibilities for the development of high performance sensors.

## 1. Introduction

The detection of explosives remains a challenge to the rapid development of modern life [1]. The detection technology of explosives requires not only simple and inexpensive constituents, but also the ability to detect specific explosives quickly and accurately. In 1895, Wolffenstein found and synthesized triacetone triperoxide (TATP) [2], which is a self-made explosive synthesized from the common chemicals acetone (C_3_H_6_O) and hydrogen peroxide (H_2_O_2_) found in daily life [3,4]. Because it is difficult to detect [5,6], it is more popular for terrorist activities [4,7,8,9,10]. Therefore, it is very important to carry out the continuous monitoring of TATP precursors (C_3_H_6_O and H_2_O_2_) in public places [11,12]. In the past, researchers have also reported many TATP detection methods. For example, infrared (IR) [13], Raman spectroscopy [14,15], and mass spectrometry (MS) [16,17] detection methods; these detection methods have some defects of low sensitivity, slow response, high production cost, and a certain risk of detection [18,19]. In recent years, many articles on the detection of H_2_O_2_ have been reported, and their applications include environmental, biological, food, and industrial fields, using such as the non-enzymatic chemi-resistive H_2_O_2_ sensor [20,21], and reports of a C_3_H_6_O sensor [22,23,24], which achieve the advantages of low cost detection. In fact, simultaneous detection of H_2_O_2_ and C_3_H_6_O would provide a viable basis for the detection of explosive TATP. Of course using electrochemical detection of TATP precursors is not a new concept [8,25,26]. The Vladimir [27] team introduced a good response with nano-spring-based sensors for gases such as TATP precursors, as well as determining conditions for In_2_O_3_ [28] and WO_3_ [29] but still did not achieve operation at room temperature. The application of graphene in chemical gas sensors is receiving more and more attention [30,31]. Some important factors affecting the sensing properties of graphene were found in the process of continuous optimization of graphene sensing performances, for example, structural defects [32] and doped metal oxides [33]. Experiments [34] have shown that the defects at the edges of graphene nanoribbons interact more strongly with gas molecules, and the reduced graphene oxide (RGO) has a high defect density for fabricating gas sensors [35]. The composite materials based on RGO and metal oxides have been widely used for gas sensing, RGO/Fe_2_O_3_ nanocomposites achieve high selectivity to NO_2_ at room temperature and RGO/SnO_2_ etc. achieve detection of ambient gases at room temperature. Based on these experiments, it is necessary to achieve high sensitivity, rapid response, and accurate identification detection of TATP precursor gas at room temperature. It can be seen that RGO-based composite materials provide hope for the sensitive detection of TATP precursors at room temperature. More recently, composite materials of RGO/MoS_2_ have been fabricated and proved to be effective in gas sensing performance. The Bon-Cheol Ku team reported that the MoS_2_/RGO composite film can be used as a highly sensitive gas sensor that can detect concentrations of harmful gases such as NO_2_ as low as 0.15 ppm [35]. It is reported that the RGO/MoS_2_ hybrid film not only shows improved sensitivity to CH_2_O at room temperature, but also exhibits fast response characteristics and good reproducibility [36].

Both RGO and MoS_2_ are typical two-dimensional nanomaterials, which are generally p-type and n-type semiconductors, respectively [37,38]. As far as we know, RGO/MoS_2_ composites have not been used to detect TATP precursors sensitively at room temperature. The synergistic effect of RGO and MoS_2_ in gas sensing detection of TATP precursors is worth studying. Inspired by the above researches, in order to detect precursors of TATP effectively, the hierarchical MoS_2_/RGO composites were synthesized by a hydrothermal method, using two-dimensional RGO as template. The effects of the ratio of RGO to raw materials for the synthesis of MoS_2_ on the morphology, structure, and gas sensing properties of the MoS_2_/RGO composites were studied.

## 2. Materials and Methods

### 2.1. Preparation of MoS_2_/RGO Composites

Ammonium molybdate [(NH_4_)_6_Mo_7_O_24_·4H_2_O], thiourea (CH_4_N_2_S), and ethanol (C_2_H_6_O), analytical reagents, were purchased from Sinopharm Chemical Reagent Co., Ltd. H_2_O_2_ (30%) was purchased from Aladdin Reagent Co., Ltd. Graphene Oxide (GO) was synthesized from natural flake graphite (100 mesh) by Hummers method [39,40]. The preparation process of the MoS_2_/RGO composites can be summarized as follows. Amounts of, 1, 0.5, 0.33 mmol (NH_4_)_6_Mo_7_O_24_·4H_2_O and 30, 15, 10 mmol CH_4_N_2_S were dissolved in 35 mL diluted graphene solution, respectively, and stirred for 30 min to make them homogeneous. The three solutions were then transferred into three 45 mL polytetrafluoroethylene (PTFE) stainless steel autoclaves and maintained at 180 °C for 24 h. As shown in Figure 1, the raw materials for synthesis of MoS_2_ adsorbed on RGO first nucleate at high temperature to form MoS_2_ nanocrystals, and then form MoS_2_/RGO composites. Finally, the reaction system was cooled to room temperature naturally, the product was collected by centrifugation, washed with deionized water, and the sample preparation was concluded after drying for 20 h at 70 °C. For the convenience of description, the MoS_2_/RGO composites from 1, 0.5, 0.33 mmol (NH_4_)_6_Mo_7_O_24_·4H_2_O and 30, 15, 10 mmol CH_4_N_2_S were designated as MoS_2_/RGO-1, MoS_2_/RGO-2 and MoS_2_/RGO-3.

### 2.2. Characterization

The crystal structure of MoS_2_/RGO was characterized by X-ray diffraction (XRD) (Bruker D8 Advance, with Cu-K_α_ radiation). The morphology of MoS_2_/RGO was observed by transmission electron microscopy (TEM, JEM-2100F, Japan) and field emission scanning electron microscopy (FE-SEM, S-4800, Hitachi, Japan). The surface properties of MoS_2_/RGO were recorded using a Fourier Transform Infrared (FT-IR) spectrometer (Bruker-V Vertex 70, Karlsruhe, Germany). Raman Detection of samples was with a Raman Spectrometer (Raman spectrometer, Horiba Company, iHR550, Shanghai, China). The chemical composition of the main elements was studied by X-ray photoelectron spectroscopy (XPS K-Alpha+, Thermo Fisher Scientific, Waltham, MA, USA). The I–V curves of the sensors were tested by an electrochemical workstation (CIMPS-2, ZAHER ENNIUM) at room temperature.

### 2.3. Manufacture and Testing of Sensor Parts

The blank sensor chip was purchased from Beijing Elite Co., Ltd., Beijing, China. Platinum interdigitated electrodes with both finger-width and interfinger spacing of about 200 μM were printed on a ceramic substrate, forming a blank sensor chip. First, the sample was mixed with a quantity of deionized water to form a uniform slurry, and then the platinum finger fork electrode used to apply the uniform slurry to the ceramic substrate while the sensing film was formed by drying at room temperature (25 °C) for 24 h. The sensors based on MoS_2_/RGO-1, MoS_2_/RGO-2, and MoS_2_/RGO-3 were designated as sensor 1, sensor 2, and sensor 3, respectively. Finally, the sensor was aged in air for about 24 h with a 0.5 V voltage to ensure good stability. The gas sensing tests, including the definition of response, response time, and recovery time in this work are similar to the previous report [41]. The specific sensing tests of H_2_O_2_, C_3_H6O, C_2_H_6_O vapors, and Ozone (O_3_) are shown in the Appendix A, and the sensitivity data was recorded by an electrochemical workstation (CIMPS-2, ZAHER ENNIUM) in a 25 °C air-conditioned room.

## 3. Results and Discussion

### 3.1. Characterization Results of MoS_2_/RGO

Figure 2a–f shows the scanning electron microscope (SEM) images of the obtained MoS_2_/RGO composites. It can be seen from the graph that the evolution of the morphology structure of the composites varies with the ratio of RGO to raw materials for the synthesis of MoS_2_. The raw materials for synthesis of MoS_2_ adsorbed on RGO first nucleate at high temperature to form MoS_2_ nanocrystals, and then form MoS_2_/RGO composites. Evidently, for the MoS_2_/RGO-1 and MoS_2_/RGO-2 composites, RGO was relatively small relative to MoS_2_, which was almost completely coated by the excess of MoS_2_ (Figure 2a–d). In the precursor mixed solution of MoS_2_/RGO-1 of MoS_2_/RGO-3, the content of RGO remained unchanged, while the ratio of ammonium molybdate and thiourea decreased gradually. For the MoS_2_/RGO-3 composites, the amount of raw materials ((NH_4_)_6_Mo_7_O_24_·4H_2_O and CH_4_N_2_S) for the synthesis of MoS_2_ is only one third of that for MoS_2_/RGO-1 composites. For the MoS_2_/RGO-1 and MoS_2_/RGO-2, the concentrations of (NH_4_)_6_Mo_7_O_24_·4H_2_O and CH_4_N_2_S are higher, so the nucleation rate and growth rate are faster than that of MoS_2_/RGO-3. As a result, MoS_2_/RGO-1 and MoS_2_/RGO-2 grow rapidly into small grains (Figure 2a–d). In contrast, the nucleation rates and growth rate of MoS_2_/RGO-3 are slower because of its lower concentration, and it grows into a finer pattern structure (Figure 2e,f). Therefore, MoS_2_ can grow well with RGO as template, and finally form the hierarchical MoS_2_/RGO-3 composites (Figure 2e,f). As shown in Figure 2e,f, the MoS_2_/RGO-3 composites eventually form interconnected, patterned spheres and the thickness of the curved pattern MoS_2_ sheet is only about 20 nm. The hierarchical structure, ultra-thin thickness provides sufficient channels and sites for the adsorption and desorption of the target gas, which is helpful to improve the sensitivity of the sensor and the speed of adsorption and desorption.

X-ray diffraction (XRD) analysis was performed on MoS_2_/RGO-1, MoS_2_/RGO-2, and MoS_2_/RGO-3 composites to examine the crystal structure. As shown in Figure 3a, the diffraction peak at 2θ = 28.5° is sharp, and the sharpness becomes larger as the ratio changes, indicating that the crystallinity of the sample is constantly improving [42]. In addition, the position of the peak is also shifted to the left, indicating that the lattice size has changed, this result is consistent with the measured results of the SEM. It is noteworthy that the absence of high-indexed diffraction peaks indicates short-range disordering nature in the products, which may offer more active sites for gas sensing [41]. To determine the functional groups contained in the samples, FT-IR analysis was performed and is shown in Figure 3b. Due to the presence of hydroxyl groups (–OH), the MoS_2_/RGO samples showed 3140 cm^−1^ and 3428 cm^−1^ peaks in the range of 3000–3800 cm^−1^, respectively [43]. The absorption peak at 1624 cm^−1^ is related to the in-plane vibration of the H–O–H bending band of the adsorbed H_2_O molecule or the C–C bonding of the sp^2^ hybrid, 1400 cm^−1^ (carboxy O–H stretching) [44], and 1031 cm^−1^ (C–O) [43]. Because RGO was not observed in the SEM images, Raman spectroscopy was performed to determine the composition of the composites (Figure 3c). The Raman spectra of the composites shown at 1305 cm^−1^ and 1542 cm^−1^ correspond to the D, G bands of RGO, respectively. The G band is produced by the surface vibration of the sp^2^ carbon atom, which is consistent with the results of the FT-IR test while the D band is usually considered to be the disordered vibration peak of graphene [45,46]. These results prove the existence of RGO in the composites. I–V characteristic curves of the sensors based on MoS_2_/RGO composites also were performed to prove the existence and function of RGO in the MoS_2_/RGO composites (Figure 3d). The linear I–V relations showed a perfect ohmic contact between MoS_2_/RGO composites and the metal electrode [47,48]. Compared with MoS_2_, RGO has the better conductivity. Therefore, with the increase of RGO proportion in the composites, the conductivity increases from MoS_2_/RGO-1 to MoS_2_/RGO-3, which also proves the existence and function of RGO in the composites. The existence of RGO was also demonstrated by direct TEM observations. As shown in Figure 3e,f, TEM images of the MoS_2_/RGO composite shows the wrinkled RGO and the MoS_2_ anchored on RGO. The MoS_2_/RGO composite provides nanoscale MoS_2_ flakes in the form of loose agglomerates on RGO, which can also be distinguished by their brightness. As shown by the yellow lines in the Figure 3f, one can directly observe the crystalline interplanar spacing attributed to MoS_2_, and the average interplanar spacing from the six spacing was calculated to be 0.63 nm. The 0.63 nm of interplanar spacing was attributed to the (002) planes of MoS_2_ [45], proving the existence of the MoS_2_ anchored on RGO. These observations are consistent with the Raman results, indicating the successful preparation of MoS_2_/RGO composites.

In order to further confirm the distribution of the elements contained in the composites, the chemical states of the MoS_2_/RGO composite were investigated by XPS. Figure 4a shows that the main constituent elements in the MoS_2_/RGO composites are S, Mo, C, and O. The high-resolution Mo3d spectrum shows two distinct peaks at 227.86 and 231.12 eV, corresponding to Mo3d_5/2_ and Mo3d_3/2_, respectively (Figure 4d). The binding energies of 160.81 and 161.9 eV correspond to S2p_3/2_ and S2p_1/2_, respectively [49] (Figure 4e). One can see intuitively from the Figure 4b–e that except for C, the corresponding peaks of S, Mo, and O are reduced from the MoS_2_/RGO-1 to MoS_2_/RGO-3, which is consistent with the increasing proportion of RGO in the composites. It is worth noting that with the increasing proportion of RGO in the composites, the intensity ratio of C1s to S2p increases gradually (Figure 4f). This normalized result is consistent with the previous characterization analysis, demonstrating that the ratio of RGO to raw materials for synthesis of MoS_2_ effectively influences the components, morphology, and structures of the MoS_2_/RGO composites. One can expect that these changes will also have a significant impact on the gas sensitivity of MoS_2_/RGO composites [50].

### 3.2. Fabrication and Testing of Sensor Array

Figure 5a shows the dynamic sensing curves of the sensors based on different samples at room temperature to 50 ppm of H_2_O_2_, C_3_H_6_O, and C_2_H_6_O vapors. As can be clearly seen from the sensing curves, the three sensors based on MoS_2_/RGO composites respond upward to oxidizing H_2_O_2_ vapor and downward to reducing C_3_H_6_O and C_2_H_6_O vapors, reflecting the sensing characteristics of p-type semiconductors. Generally, RGO and MoS_2_ are p-type and n-type semiconductors, respectively. The p-type sensing characteristics prove that RGO plays an important role in gas sensing of MoS_2_/RGO composites. It is worth noting that the responses of the three sensors based on MoS_2_/RGO composites to 50 ppm of H_2_O_2_, C_3_H_6_O, and C_2_H_6_O vapors increases with the increase of RGO content (Figure 5b). The responses to 50 ppm of H_2_O_2_ vapor increased from 29.0% to 59.6%, and then to 373.1% for the sensors 1, 2, and 3, respectively. This trend of responses also applies to 50 ppm of C_2_H_6_O vapor, but not C_3_H_6_O vapor. This phenomenon can be attributed to the charge depletion layer and hierarchical structures. It is well known that both the charge depletion layer and the particle size of the semiconductor materials determine the sensing performance of the chemi-resitive sensor. The higher the proportion of the electron depletion layer in the semiconductor particle, the better the gas sensitivity of the sensing materials. The MoS_2_/RGO-3 composite has a pattern-like hierarchical structure with a thickness of about 20 nm, and target gas molecules can be adsorbed on both sides of the sheet structures to form a deeper charge depletion layer. In addition, the hierarchical structure of MoS_2_/RGO-3 composite provides a larger specific surface area and more active sites. As a result, the deeper charge depletion layer, larger specific surface area, and more active sites of the MoS_2_/RGO-3 composite contributed to the higher sensitivity, which is consistent with the results of the gas sensitivity test. In contrast, pure MoS_2_ was also prepared, and their two-dimensional sheet morphologies are shown in Appendix A. The sensing curves of sensors based on pure MoS_2_ and RGO to 1000 ppm of H_2_O_2_, C_3_H_6_O, and C_2_H_6_O vapors were also tested (Appendix Aa). It can be seen from the Appendix A that although the concentration of H_2_O_2_, C_3_H_6_O, and C_2_H_6_O vapors increased 20 times (1000 ppm), the corresponding responses of the sensors based on pure MoS_2_ and RGO hardly increased significantly compared with the responses of the sensors based on MoS_2_/RGO composites to 50 ppm of H_2_O_2_, C_3_H_6_O, and C_2_H_6_O vapors. This fully illustrates the important role of the synergistic effect of MoS_2_ and RGO in gas sensing. In addition, for the three target gases, all three sensors based on MoS_2_/RGO composites show very short response time and recovery time, reflecting the fast adsorption and desorption of target gases on the surface of MoS_2_/RGO composites. As shown in Figure 5c, the maximum response time and recovery time are no more than 14.5 s and 16.3 s, respectively, proving the real-time sensing performance of the sensors. For an excellent gas sensor, not only is high response required in practical applications, but also good selectivity to the target gas. C_2_H_6_O is a common volatile organic compound that interferes greatly with the detection of TATP precursors, it is very necessary to use C_2_H_6_O vapor as an interference factor in the detection of TATP precursors. Unfortunately, the MoS_2_/RGO-3 composite has a higher response to C_2_H_6_O vapor than that of C_3_H_6_O vapor (Figure 5a,b), showing the poor anti-interference characteristics to C_2_H_6_O vapor. Moreover, O_3_ is a strong oxidizing gas in the air and O_3_ often interferes with the sensing detection of oxidizing gases. Therefore, we also tested the sensors based on MoS_2_/RGO composites to 50 ppm of O_3_ gas (Appendix A). It can be seen from Appendix A that the sensors based on MoS_2_/RGO composites hardly respond to 50 ppm of O_3_, showing good anti-interference ability of O_3_.

The sensing performances of sensors based on MoS_2_/RGO-3 and other reported chemi-resistive sensors for detection of H_2_O_2_ vapor can be found in Table 1. As can be seen clearly from the Table 1, our sensors also work at room temperature like the reported sensors, but the sensor in our work has the highest sensitivity to H_2_O_2_ vapor at room temperature, achieving a response of 373.1% to 50 ppm H_2_O_2_ vapor. The response time for 50 ppm H_2_O_2_ vapor is approximately 9 s. This comparison shows that our sensor of MoS_2_/RGO-3 has better comprehensive sensing performance for H_2_O_2_ vapor.

In addition, the response of the MoS_2_/RGO-3 sensor to different concentrations of H_2_O_2_ vapor was tested (Figure 6a). Based on the results, the estimated detection limit (defined as the detection limit = 3S_D_/m, where m is the slope of the linear portion of the calibration curve, S_D_ is the standard deviation of the noise in the response curve) [41], and the H_2_O_2_ vapor is determined to be 0.65 ppm (Figure 6b). The results show that MoS_2_/RGO-3 composites have potential applications of gas detection of TATP precursors.

### 3.3. Discriminative Capability of the Sensor Array

To evaluate the discriminative capability of a simple array consisting of three sensors, all the responses were further analyzed using a principal component analysis (PCA) method and radar method combining kinetic and thermodynamic parameters. The kinetic and thermodynamic parameters of the interaction of the analytes and the sensor array are utilized to assess the discriminative capability of the sensor array. The radar method refers to fingerprint recognition, which creates a unique database of explosive fingerprints, enabling the separation of similar chemical entities and providing a fast and reliable method for identifying individual chemical reagents [54,55]. The responses and response time inherent in the interaction between each analyte and the three sensors were chosen as kinematic and thermodynamic parameters, respectively. Therefore, its three pairs of sensing responses and response times from the sensor array were used to calculate the ratio of responses to response times, and the three parameters obtained for each analyte represent the fingerprint. From the fingerprints obtained (Figure 7a–c), one can see that the triangular fingerprints corresponding to H_2_O_2_, C_3_H_6_O, and C_2_H_6_O vapors are not well differentiated because the number of sensors is too small. Therefore, the PCA method is used to evaluate the discriminative ability of the sensor array. PCA is a popular multivariate statistical technique used to simplify data sets. The purpose of this method is to reduce the dimension of multivariate data while retaining as much relevant information as possible [56,57]. Data sets of principal component analysis applied to pattern recognition and/or gas recognition have been reported [58]. As shown in Figure 7d, it can be clearly seen that the simple sensor array is very effective in distinguishing three target analytes, showing the discriminative capability. It also proves that the design of gas sensing properties of MoS_2_/RGO composites by changing the ratio of RGO to MoS_2_ is effective and feasible. Considering that the maximum response time of the sensor array is just 14.5 s, means that the simple sensor array can detect three analytes in 14.5 s.

### 3.4. Analysis of the Possible Sensing Mechanism

The conductivity of the sensing material depends on the adsorbed gas molecules (oxidizing or reducing) [59]. Generally, MoS_2_ acts as an n-type semiconductor [37,38], while RGO is considered to be a p-type semiconductor with a defect site and a functionalized group on its surface, which acts as an active site for the gas, facilitating the adsorption of gas molecules [60]. The gas sensing results show that the MoS_2_/RGO composite exhibits the characteristics of p-type semiconductors. When the sensors based on MoS_2_/RGO composites were exposed to the reducing C_3_H_6_O vapor, the following reactions occurred.
C_3_H_6_O (g) + 4O_2_^−^ (s) → 3CO_2_ + 3H_2_O + 4e^−^(1)

According to Equation (1), MoS_2_/RGO composites captured the electrons from the reducing C_3_H_6_O vapor, while the conductivity of composites decreases, exhibiting characteristics of p-type semiconductors. It is reported that H_2_O_2_ will react in the following two ways depending on the concentration of hydrogen peroxide. At high H_2_O_2_ (of about 10 vol%) concentrations the mechanism is as follows [61]:2H_2_O_2_ → 2H_2_O + O_2_(2)

At lower concentrations (2.1 vol%) the net reaction is:2H_2_O_2_ → 2H_2_O + 0.87O_2_ + 0.08O_3_(3)

In our work, H_2_O_2_ with a mass fraction of 30% was used. According to Equation (2), the main product of H_2_O_2_ decomposition is O_2_. Therefore, the produced O_2_ and H_2_O_2_ vapor will capture electrons from the MoS_2_/RGO composites, and the conductivity of the composites increases, also exhibiting characteristics of p-type semiconductors. This indicated that RGO played an important role in the gas sensing properties of MoS_2_/RGO composites. Because RGO/MoS_2_ conjugates can form excellent charge transfer pathways [45], the charge can favorably travel from MoS_2_ to RGO quickly, resulting in a very large and fast variation of the conductivity. This synergy of MoS_2_ and RGO contributes to a quick and sensitive response to target analytes. With the decrease of MoS_2_ (or the increase of RGO) in the RGO/MoS_2_ composite, the contact between RGO and MoS_2_ is more sufficient, which is more conducive to giving a good gas sensing performance. Therefore, the sensitivity of the sensor based on MoS_2_/RGO composites was significantly improved with the increase of RGO within an appropriate range. Without doubt, the formation of hierarchical structure of the MoS_2_/RGO composites is also conducive to improving sensitivity, which is consistent with the very good sensitivity of sensor-3 of MoS_2_/RGO composites.

## 4. Conclusions

p-type RGO and n-type MoS_2_, typical two-dimensional nanomaterials, were used successfully to design hierarchical MoS_2_/RGO composites using RGO as templates. The effects of the ratio of RGO to raw materials for the synthesis of MoS_2_ on the morphology, structure, and gas sensing properties of the MoS_2_/RGO composites were studied in order to detect the precursors of TATP effectively. It was found that after optimization, the response to 50 ppm of H_2_O_2_ vapor was increased from 29.0% to 373.1%, achieving an increase of about 12 times. Meanwhile, all three sensors based on MoS_2_/RGO composites exhibited excellent anti-interference performance to ozone with strong oxidation. Furthermore, the simple sensor array based on MoS_2_/RGO composites achieved discriminative detection of three target analytes in 14.5 s at room temperature. This proves that the design of gas sensing properties of MoS_2_/RGO composites by changing the ratio of RGO to MoS_2_ is effective and feasible. The synergistic effect between two-dimensional RGO and MoS_2_ provide new possibilities for the development of high performance sensors.

## Figures and Tables

**Figure 1 sensors-19-01281-f001:**
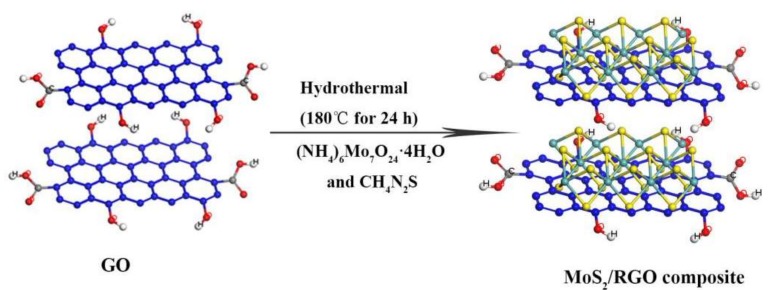
Schematic diagram of MoS_2_/RGO composite with reduced graphene oxide (RGO) as template.

**Figure 2 sensors-19-01281-f002:**
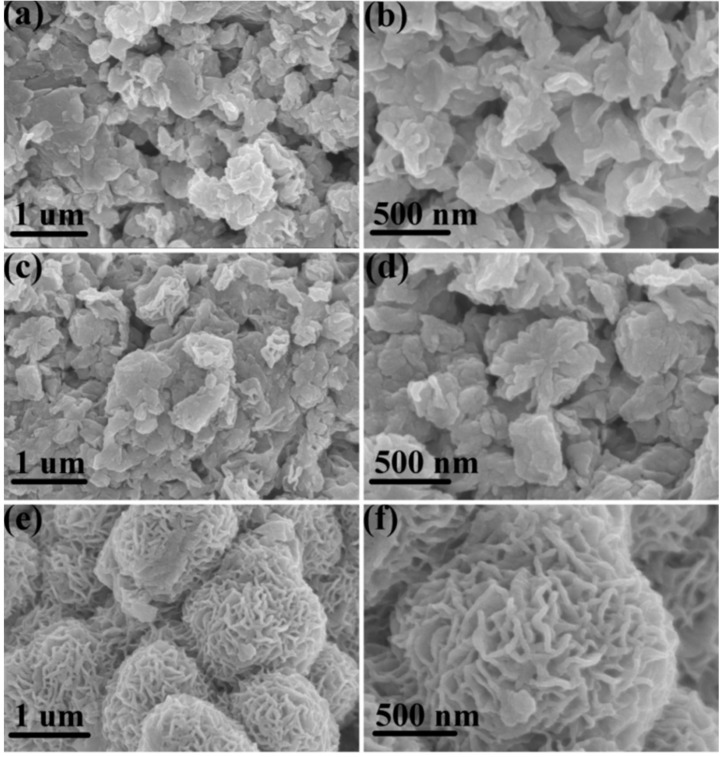
SEM images of (**a**,**b**) MoS_2_/RGO-1, (**c**,**d**) MoS_2_/RGO-2, and (**e**,**f**) MoS_2_/RGO-3 composites.

**Figure 3 sensors-19-01281-f003:**
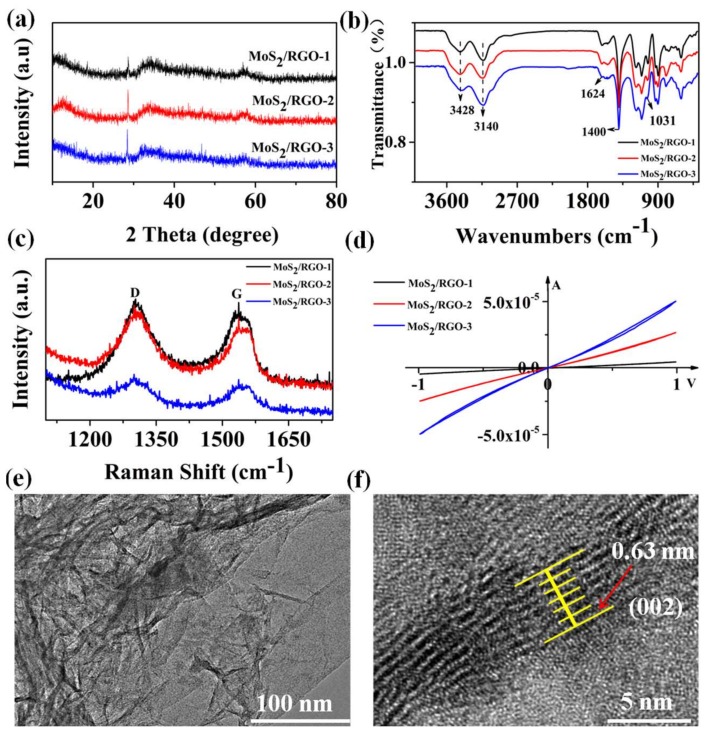
(**a**) XRD patterns, (**b**) FT-IR spectrum, (**c**) Raman spectrum of MoS_2_/RGO-1, MoS_2_/RGO-2, and MoS_2_/RGO-3 composites; (**d**) I–V curves of sensors 1, 2, 3 and (**e**,**f**) TEM images of MoS_2_/RGO-3 composites.

**Figure 4 sensors-19-01281-f004:**
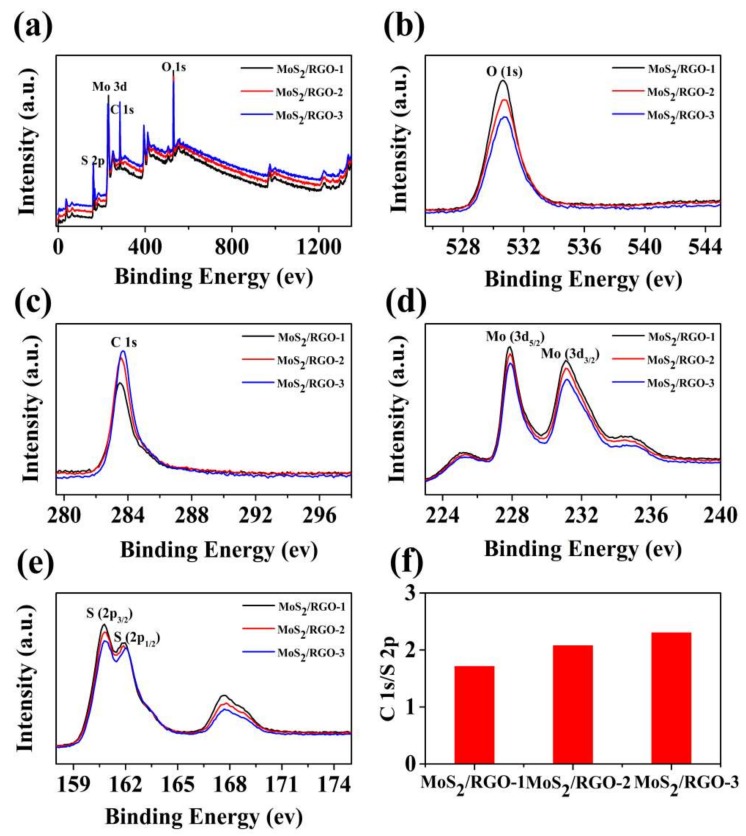
XPS spectra of MoS_2_/RGO composites (**a**) high-resolution spectra, (**b**) O1s (**c**) C1s, (**d**) Mo3d, (**e**) S2p, and (**f**) ratio of the intensity of C1s to S2p.

**Figure 5 sensors-19-01281-f005:**
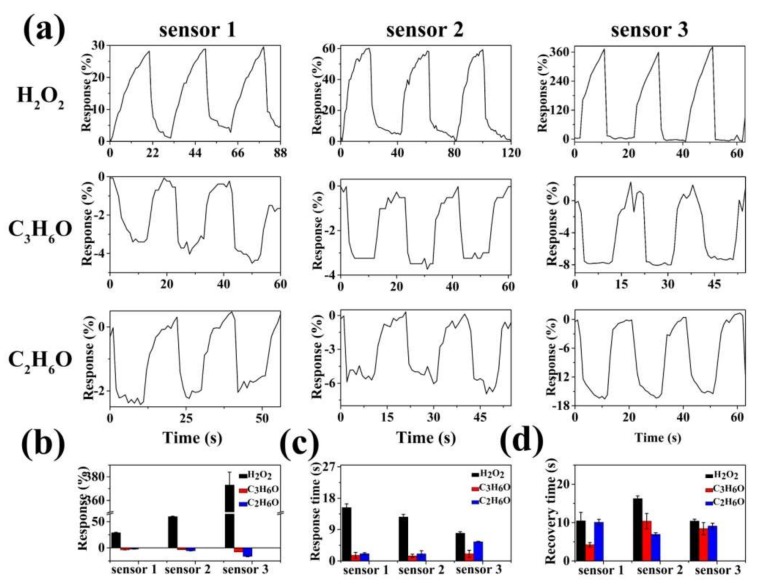
(**a**) Dynamic sensing curves of the devices based on sensor 1, 2, and 3 to 50 ppm of H_2_O_2_, C_3_H_6_O, C_2_H_6_O vapors at room temperature; statistical graph of (**b**) average response, (**c**) response time and (**d**) recovery time corresponding to the sensing curves.

**Figure 6 sensors-19-01281-f006:**
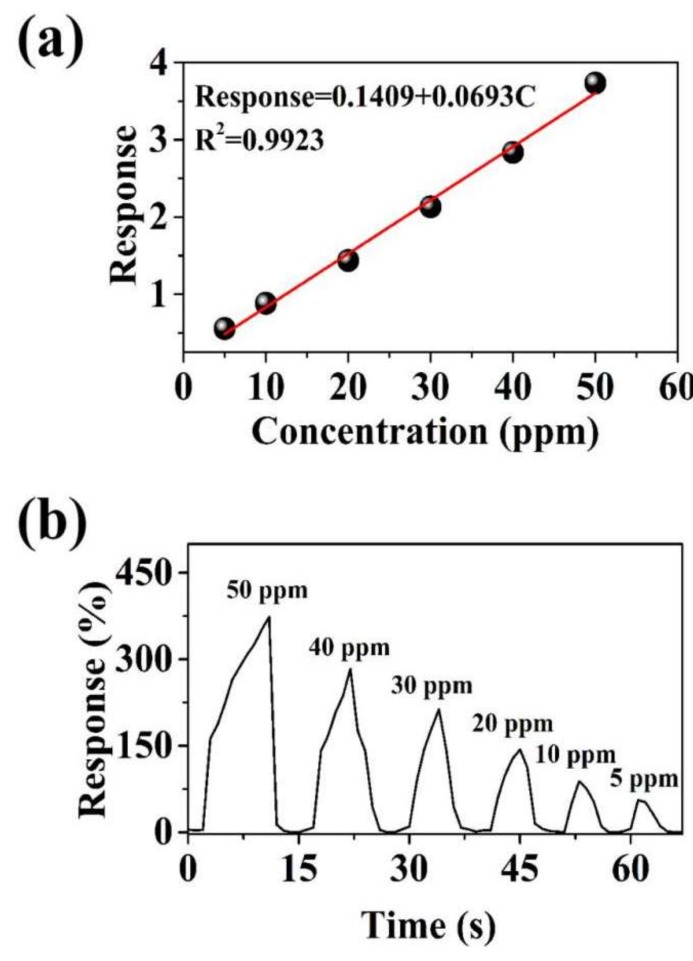
(**a**) Response curves of MoS_2_/RGO-3 sensor to different concentrations of H_2_O_2_ vapor and (**b**) plots of the fitting of response vs concentration.

**Figure 7 sensors-19-01281-f007:**
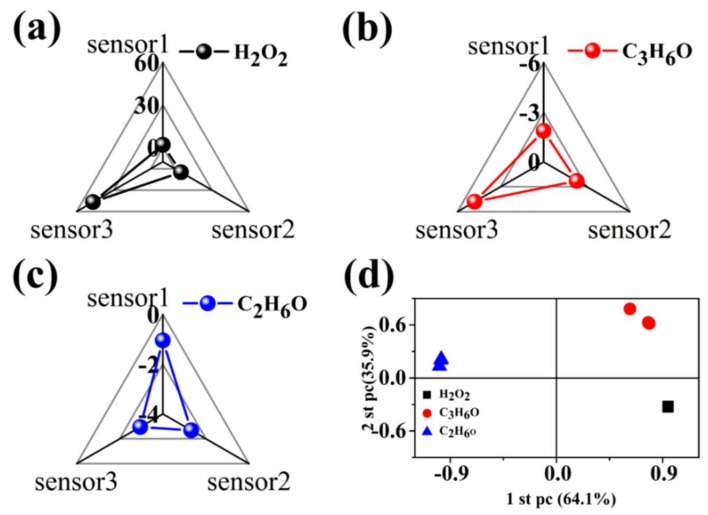
Fingerprints combining kinetic and thermodynamic parameters of (**a**) H_2_O_2_, (**b**) C_3_H_6_O, (**c**) C_2_H_6_O; (**d**) Two-dimensional principal component analysis (PCA) plots according to the responses to H_2_O_2_, C_3_H_6_O, and C_2_H_6_O of sensor array.

**Table 1 sensors-19-01281-t001:** Comparison of the reported H_2_O_2_ sensors and our sensors of MoS_2_/RGO-3.

Sensing Materials	Analyte Concentration	Temperature (°C)	Response (%)	Response Time	Ref.
SWCNTs	100 ppm H_2_O_2_	25	~1.97	~20 s	[51]
Pt-SWCNTs	60.6 ppm H_2_O_2_	23 ± 1	~50	~240 s	[52]
CuPc-f-MWNTs CoPc-f-MWNTs VPc-f-MWNTs	34% H_2_O_2_ (aq) vapors	Ambient temperature	(−) 24.2 (+) 3.8 (−) 4.4	2 s 2 s 4 s	[53]
MoS_2_/RGO-3	50 ppm H_2_O_2_	25	~373.1	~9 s	This work

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
