# Peer review of "Detection of Triacetone Triperoxide (TATP) Precursors with an Array of Sensors Based on MoS2/RGO Composites"

_sensors, 2019, doi:10.3390/s19061281_

Round 1
Reviewer 1 Report
This article details the design and testing of a MoS2-RGO composite material for the detection of vapors from chemicals used for the manufacture of TATP, an explosive. The authors detailed the construction, provide performance results, and suggest a reasonable mechanism for the responses seen by the sensors. Overall, this was an interesting paper and should find favor with the analytical sensing community. However, the overall style of the paper would greatly improve with further editing for English. This reviewer does recognize that English is not the first language of the authors, but there are certain stylistic and editing improvements that can be made to make this article clearer. A few of these will be pointed out, but additional editing is necessary. Overall, the recommendation from this reviewer is to publish with some changes that will be addressed below:
1. Overall, the supporting information section is fine and germane to the article. No changes are needed.
2. Page 1, Abstract, Line 14: the authors state “…common chemical in self-life of acetone…” Should this be shelf-life?
3. Page 1, Line 35: the word chemical is repeated
4, Page 2, Line 78: GO should probably be identified as Graphene Oxide. The abbreviation has not been used before this occurrence.
5. Page 3, Sections 2.2 and 2.3: It is suggested that the order of these two sections should be reversed. In other words, Section 2.2 should be Characterization, and Section 2.3 should be Manufacture and testing of sensor parts. This is the order that the Results are presented.
6. Section 2.2 (Manufacture and testing of sensor parts): This section should be expanded in regards to the electrode design. While the authors state that a platinum finger fork electrode is used, it is not really clear its details. The authors should provide information on whether these electrodes were purchased or made in house. Additionally, details of the electrode design would be appreciated, including electrode spacing, length, etc. If these electrodes were made in-house, a sentence or two describing the general design would be appreciated (such as how the platinum was applied, thickness, etc.).
7. Page 6, Line 152: Should “cures” be “curves”?
8. Section 3.2 (Fabrication and testing of sensor array): It is understood that both H2O2 and C3H6O are precursors for the synthesis of TATP, but why is C2H6O being tested? In addition, is this dimethyl ether or ethanol (I think it is dimethyl ether, just want to be sure)? Also, the response of the sensors show a positive response to H2O2 (oxidizing) and negative responses (reducing) to both C3H6O and C2H6O. Is there a way to discriminate between the two different responses to determine if both the oxidizing and reducing components are present? That is most likely what the small sensing array is needed for, but it is not really clear.
9. Page 8, Lines 222-224: It might be prudent to add a general reference describing the PCA and radar methods for analyzing this data. This would be useful for those not well versed in array and neural net analysis techniques.
10. Page 8, Figure 6: a short description of part (d) is needed.
Author Response
Reviewer 1's comments to the Author
This article details the design and testing of a MoS2-RGO composite material for the detection of vapors from chemicals used for the manufacture of TATP, an explosive. The authors detailed the construction, provide performance results, and suggest a reasonable mechanism for the responses seen by the sensors. Overall, this was an interesting paper and should find favor with the analytical sensing community. However, the overall style of the paper would greatly improve with further editing for English. This reviewer does recognize that English is not the first language of the authors, but there are certain stylistic and editing improvements that can be made to make this article clearer. A few of these will be pointed out, but additional editing is necessary. Overall, the recommendation from this reviewer is to publish with some changes that will be addressed below:
Response to Reviewer 1's Comments
Point 1: Overall, the supporting information section is fine and germane to the article. No changes are needed.
Response 1: Thank you very much for your affirmation.
Point 2: Page 1, Abstract, Line 14: the authors state “…common chemical in self-life of acetone…” Should this be shelf-life?
Response 2: Thank you very much for your reminder. This is due to improper wording. We have changed Page 1, Abstract, Lines 14-15: to “Triacetone trioxide (TATP) is a self-made explosive synthesized from the commonly used chemical acetone (C2H6O) and hydrogen peroxide (H2O2).
Point 3: Page 1, Line 35: the word chemical is repeated.
Response 3: Thank you very much for your kind reminder. We have deleted the repeated word “chemical” in Line 35 on Page 1.
Point 4: Page 2, Line 78: GO should probably be identified as Graphene Oxide. The abbreviation has not been used before this occurrence.
Response 4: It is our negligence and we are sorry about this. So we wrote the full name of Graphene Oxide in Line 87 on Page 2, and abbreviated it in parentheses.
Point 5: Page 3, Sections 2.2 and 2.3: It is suggested that the order of these two sections should be reversed. In other words, Section 2.2 should be Characterization, and Section 2.3 should be Manufacture and testing of sensor parts. This is the order that the Results are presented.
Response 5: Thank you very much for your suggestion. In order to optimize the structure of the article, we changed Section 2.2 to Characterization and Section 2.3 to Manufacture and testing of sensor parts according to your valuable suggestions. (Lines 102-140 on Page 3 and Page 4)
Point 6: Section 2.2 (Manufacture and testing of sensor parts): This section should be expanded in regards to the electrode design. While the authors state that a platinum finger fork electrode is used, it is not really clear its details. The authors should provide information on whether these electrodes were purchased or made in house. Additionally, details of the electrode design would be appreciated, including electrode spacing, length, etc. If these electrodes were made in-house, a sentence or two describing the general design would be appreciated (such as how the platinum was applied, thickness, etc.).
Response 6: Thank you very much for your suggestion. According to the suggestion, we have clearly described the details of the platinum fork electrode purchased. The details are as follows: The blank sensor chip was purchased from Beijing Elite Co., Ltd. Platinum interdigitated electrodes with both finger-width and interfinger spacing of about 200 μm were printed on a ceramic substrate, forming a blank sensor chip. The corresponding expression is added to the experimental part (Lines 129-131 on Page 3).
Point 7: Page 6, Line 152: Should “cures” be “curves”?
Response 7: Thank you very much for reminding us of this mistake. We have changed the "cures" in Line 167 on Page 5 to "curves".
Point 8: Section 3.2 (Fabrication and testing of sensor array): It is understood that both H2O2 and C3H6O are precursors for the synthesis of TATP, but why is C2H6O being tested? In addition, is this dimethyl ether or ethanol (I think it is dimethyl ether, just want to be sure)? Also, the response of the sensors show a positive response to H2O2 (oxidizing) and negative responses (reducing) to both C3H6O and C2H6O. Is there a way to discriminate between the two different responses to determine if both the oxidizing and reducing components are present? That is most likely what the small sensing array is needed for, but it is not really clear.
Response 8: Thank you very much for your suggestion. First, we apologize for the unclear statement and we want to explain that C2H6O in this paper is ethanol. Ethanol was purchased from Sinopharm Chemical Reagent Co., Ltd, and this supplement has been added to the manuscript (Line 85 on Page2). Second, the main reasons for testing C2H6O vapor in this paper are as follows: C2H6O is a common volatile organic compound that interferes greatly with the detection of TATP precursors, it is very necessary to use C2H6O vapor as an interference factor in the detection of TATP precursors. In addition, C2H6O as an interfering factor in the detection of TATP precursors has been reported. Therefore, we detected C2H6O vapor in this work in order to make the test results more convincing. Finally, the question that both the oxidizing and reducing components are present is interesting. On the one hand, we have also considered testing the mixed vapors of oxidizing and reducing gases at the same time, but there may be an explosion in the mixed state, which poses a safety hazard. On the other hand, we can see that our sensors are not sensitive to reducing alcohol and acetone vapor, so we did not test their sensitivity to the mixed vapors of oxidizing and reducing gases. Your question points out the direction for our future research, and we will focus on this kind of problem in future research. Thank you again.
Point 9: Page 8, Lines 222-224: It might be prudent to add a general reference describing the PCA and radar methods for analyzing this data. This would be useful for those not well versed in array and neural net analysis techniques.
Response 9: Thank you very much for your valuable advice, based on your suggestion. We describe the general reference method of PCA and radar, and cite relevant literature for better understanding by those not well versed in array and neural net analysis techniques. The changes are as follows in the article: PCA is a popular multivariate statistical technique used to simplify data sets. The purpose of this method is to reduce the dimension of multivariate data while retaining as much relevant information as possible. Data sets of PCA applied to pattern recognition and/or gas recognition have been reported.(Lines 301-304 on Page 10) Radar method refers to fingerprint recognition, which creates a unique database of fingerprints belonging to analytics, enabling the separation of similar chemical entities and providing a fast and reliable method for identifying individual chemical reagents (Lines 291-293 on Page 10).
Point 10: Page 8, Figure 6: a short description of part (d) is needed.
Response 10: Thank you very much for your suggestion, under your valuable suggestions, our manuscript has been improved, the original Figure 6 in the manuscript has become Figure 7, the details of this part are as follows: PCA is a popular multivariate statistical technique used to simplify data sets. The purpose of this method is to reduce the dimension of multivariate data while retaining as much relevant information as possible. Data sets of principal component analysis applied to pattern recognition and/or gas recognition have been reported. As shown in Figure 7d, it is clearly shown that the simple sensor array are very effective in distinguishing three target analytes, showing the discriminative capability. It also proves that the design of gas sensing properties of RGO/MoS2 composites by changing the ratio of RGO to MoS2 is effective and feasible (Lines 301-309 on Page 10).
Thank you once again for spending your precious time carefully reviewing our manuscripts. Your valuable suggestions will further improve and optimize our manuscripts.
Reviewer 2 Report
The comment was attached.

Author Response
Reviewer 2's comments to the Author
In this work, Sun et al. synthesized MoS2/RGO composites with various constructors for detecting triacetone triperoxide (TATP) precursors. The proposed material is interesting to the researchers working in the field. The results seem to be reasonable, however, some statements mentioned in the manuscript are not clear enough and require more clarification. This manuscript can be considered for publication in Sensor with Major Revision after properly addressing the following issues:
Response to Reviewer 2's Comments
Point 1: In the SEM image, MoS2/rGO-3 composites showed well-organized nanostructure. It is interesting to understand the reason and formation mechanism behind. Please add some discussion about the formation mechanism of MoS2/rGO composites with various ratio.
Response 1: Thank you very much for your valuable comments. Based on your valuable suggestions, we have supplemented the formation mechanism of different proportions of MoS2/RGO composites. The details are as follows: In the precursor mixed solution of MoS2/RGO-1 to MoS2/RGO-3, the content of RGO remains unchanged, while the ratio of ammonium molybdate and thiourea decreases gradually.(Lines 151-153 on Page 4) The content of ammonium molybdate and thiourea in MoS2/RGO-3 precursor is one third of that of MoS2/RGO-1 precursor. For the MoS2/RGO-1 and MoS2/RGO-2, the concentration of ammonium molybdate and thiourea is higher, so the nucleation rate and growth rate are faster than that of MoS2/RGO-3. As a result, MoS2/RGO-1 and MoS2/RGO-2 grow rapidly into small grains (Figure 2a-d). In contrast, the nucleation rate and growth rate of MoS2/RGO-3 are slower because of its lower concentration, and it can grow into more fine pattern structure (Figure 2e, f).(Lines 154-159 on Page 5.)
Point 2: rGO or GO? It was quite confusing since there are two types of abbreviation, namely MoS2/rGO and MoS2/GO, for presenting the proposed composites.
Response 2: We are very sorry for the confusion caused by the inconsistent names of the composites in this manuscript. In this work, the representation of the composites is MoS2/RGO and we have unified the representation of the composites in the manuscript into MoS2-RGO.
Point 3: MoS2/rGO-3 composites exhibited better performance for detecting the TATP precursor gases. Why? Some of indeed analysis and discussion should be included.
a. Response 3: Thank you very much for your proposal. Based on your suggestions, we analyzed the advantages of MoS2/RGO-3 composites for the detection of TATP precursor gases. The mechanism analysis was supplemented in the revised manuscript as follows: First, MoS2/RGO-3 has a pattern-like hierarchical structure, which has better permeability and larger specific surface area than MoS2/RGO-1 and MoS2/RGO-2, and is conducive to the adsorption and desorption of target gas molecules. Second, it is well known that both the charge depletion layer and the particle size of the semiconductor materials determine the sensing performance of the chemiresitive sensor. The higher the proportion of electron depletion layer in semiconductor particle size, the better the gas sensitivity of sensing materials. As can be seen from Figure 2, MoS2/RGO-3 has a pattern-like hierarchical structure with a thickness of about 20 nm. Target gas molecules can be adsorbed on both sides of the sheet structure to form deeper electron depletion layer. As a result, the proportion of electron depletion layer is larger in the MoS2/RGO-3, resulting in the higher sensitivity.(Lines 234-244 on Page 8)
Point 4: Electrochemical impedance spectroscopy may provide some information about charge transfer resistant between the interface of electrode and gas.
Response 4: Thank you very much for your valuable suggestion. We don't know much about the information provided by electrochemical impedance spectroscopy and have consulted the relevant literature under the suggestion. Unfortunately, we still haven't understood how to analyze the information provided by electrochemical impedance spectroscopy in a short time. Under the third suggestion, we supplement the gas sensing mechanism in our revised manuscript. With your suggestion, we have learned more methods to analyze the gas sensing mechanism. We will study electrochemical impedance spectroscopy more deeply in the future, so as to make our research more systematic and in-depth.
Point 5: Please point out the detection range and LOD of MoS2/rGO composites for sensing TATP precursor gases.
Response 5: Thank you very much for your suggestion. With your valuable suggestions, we have made up the experiments to test the response of MoS2/RGO-3 composite to H2O2 vapor in the range of 50-5ppm. The concentration and response curves were fitted, and the theoretical detection limit was 0.65 ppm. The results show that MoS2/RGO-3 composites have potential applications in gas detection of TATP precursors (Lines 273-281 on Page 9).
Point 6: Interference test is also essential for characterizing the selectivity of proposed MoS2/rGO composites for sensing TATP precursor gases.
Response 6: Thanks for the suggestion. Our idea is in line with yours and in our research work we test two types of interfering gases. Reductive ethanol (C2H6O) and oxidative ozone (O3) were tested as interfering gases of acetone (C3H6O) and hydrogen peroxide (H2O2), respectively. C2H6O is a common volatile organic compound that interferes greatly with the detection of TATP precursors. It is meaningful to use it as an interfering factor in the detection of TATP precursors. The MoS2/RGO-3 composite has the higher response to C2H6O vapor that of C3H6O vapor (Figure 5b), showing the poor anti-interference characteristics to C2H6O vapor. In addition, O3 often interferes with the sensing detection of oxidizing gases. Therefore, we also tested the sensors based on MoS2/RGO composites to 50 ppm of O3 gas (Figure S4). It can be seen from the Figure S4 that the sensors based on MoS2/RGO composites hardly respond to 50 ppm of O3, showing good anti-interference ability to O3. The experimental results show that our sensor has high sensitivity to H2O2 vapor, high anti-interference ability to oxidizing gas, and low sensitivity to C3H6O vapor, poor anti-interference ability to reducing gas. The corresponding discussion was supplemented in the revised manuscript (Lines 257-265 on Page 8).
Thank you once again for spending your precious time carefully reviewing our manuscripts. Your valuable suggestions will further improve and optimize our manuscripts.
Reviewer 3 Report
The authors described a gas sensor for detection of acetone, ethanol, and hydrogen peroxide based on MoS2/rGO composites. This sensor showed rapid response time for detection of all gases. The data for the synthesis of MoS2/rGO composites shown is informative, however there are a few points for the sensing part that could be included to further improve the quality of the manuscript. I would therefore consider it favorable for publication in Sensors under the provision that the following concerns are adequately addressed.
1. As stated in the manuscript, acetone and hydrogen peroxide are the precursor of TATP. Thus, it is reasonable to monitor acetone and hydrogen peroxide vapors. However, why the authors monitor C2H6O? Is it also a precursor of TATP? There is no description of C2H6O in the introduction of the manuscript.
2.In this manuscript, C2H6O refers to ethanol or methoxymethane?
3.For the audiences of Sensors, in addition to the material characteristics, I believe they would like to imperatively know the sensor performance such detection limit. For example, the airborne permissible exposure limit for hydrogen peroxide (H2O2) is 1 ppm. Thus, 50 ppm of H2O2 is the very high concentration that may be may be harmful to human health. The author should test the sensitivity of this sensor using different concentrations of H2O2 to demonstrate the potential practical application of this sensor for H2O2.
4. For most gas sensors, two major issues for gas identification are the drift and non-selectivity of the sensors. In this manuscript, the selectivity of this sensor were demonstrated by using O3, but the gas-induced the baseline drift could be found in your sensor system. The author should discuss this baseline drift response and provide the resolution in the manuscript.
5. Please make a comparison of H2O2 sensing response of this work with those reported in the other literature.
6. Please check the grammar carefully. For example, (lane 50) ”With the continuous efforts to optimize the sensing performance of graphene,……, such as the structural defect[32] doped metal oxide[33] and Experiment[34] both clarified that the dangling bond defects at the edge of the graphene nanoribbon are more strongly interacting with the gas molecules…”
Author Response
Reviewer 3's comments to the Author
The authors described a gas sensor for detection of acetone, ethanol, and hydrogen peroxide based on MoS2/rGO composites. This sensor showed rapid response time for detection of all gases. The data for the synthesis of MoS2/rGO composites shown is informative, however there are a few points for the sensing part that could be included to further improve the quality of the manuscript. I would therefore consider it favorable for publication in Sensors under the provision that the following concerns are adequately addressed.
Response to Reviewer 3's Comments
Point 1: As stated in the manuscript, acetone and hydrogen peroxide are the precursor of TATP. Thus, it is reasonable to monitor acetone and hydrogen peroxide vapors. However, why the authors monitor C2H6O? Is it also a precursor of TATP? There is no description of C2H6O in the introduction of the manuscript.
Response 1: Thank you very much for your questions and suggestions. The main reasons for testing ethanol (C2H6O) vapor in this paper are as follows: C2H6O is a common volatile organic compound that interferes greatly with the detection of TATP precursors, it is very necessary to use C2H6O as an interference factor in the detection of TATP precursors. Therefore, C2H6O is not the precursor of TATP, it is one of the interfering factors for detecting TATP precursor in this paper.(Lines 257-259 on Page 8)
Point 2: In this manuscript, C2H6O refers to ethanol or methoxymethane?
Response 2: I am very sorry, because we did not explain the C2H6O in the manuscript, it caused reader such confusion. C2H6O in this manuscript is ethanol, and it was purchased from Sinopharm Chemical Reagent Co., Ltd. We will put the specific instructions for ethanol in the revised manuscript (Line 85 on Page 2).
Point 3: For the audiences of Sensors, in addition to the material characteristics, I believe they would like to imperatively know the sensor performance such detection limit. For example, the airborne permissible exposure limit for hydrogen peroxide (H2O2) is 1 ppm. Thus, 50 ppm of H2O2 is the very high concentration that may be may be harmful to human health. The author should test the sensitivity of this sensor using different concentrations of H2O2 to demonstrate the potential practical application of this sensor for H2O2.
Response 3: Thank you very much for your suggestion, your suggestion is very valuable and meaningful. With your valuable suggestions, we have made up the experiment of to test the response of MoS2/RGO-3 composite to H2O2 vapor in the range of 50-5ppm. The concentration and response curves were fitted, and the theoretical detection limit was 0.65 ppm. The results show that MoS2/RGO-3 composites have potential applications in gas detection of TATP precursors (Lines 273-281 on Page 9).
Point 4: For most gas sensors, two major issues for gas identification are the drift and non-selectivity of the sensors. In this manuscript, the selectivity of this sensor was demonstrated by using O3, but the gas-induced the baseline drift could be found in your sensor system. The author should discuss this baseline drift response and provide the resolution in the manuscript.
Response 4: Thank you very much for your suggestion. In this manuscript, we used O3 and C2H6O vapor to prove the selectivity of the sensor, but from the response we found that there is a baseline drift problem for sensors 1 and 2. The possible cause of baseline drift is that the composite materials MoS2/RGO-1 and MoS2/RGO-2 grow into large-sized blocks, while MoS2/RGO-3 grows into a regular pattern with a thickness of 20 nm, the large specific surface area of MoS2/RGO-3 can provide more adsorption sites and facilitate desorption, The large particle bulk composites MoS2/RGO-1 and 2 do not have such advantages. Therefore, sensor 3 is more sensitive than sensors 1 and 2, and there is substantially no baseline drift problem. A possible cause of a large drift in the O3 baseline is that O3 is a strong oxidizing gas, after the sensor adsorbs O3, the desorption process is slow, and the desorption process is not complete, which will cause the baseline to drift. For solutions that encounter baseline drift, the desorption process can be thoroughly achieved by increasing the time of desorption, but this will increase the recovery time of the sensor. Or it is possible to accelerate the desorption process of gas by ultraviolet irradiation. This method will increase the cost of production. Therefore, it is the goal of scientific research workers to develop low-cost, fast-responding gas sensors. In addition, considering the overall structure of the manuscript, after careful consideration, we believe that it is more appropriate to put the O3 baseline drift part of the description in the support information. Corresponding to Figure S4, it is convenient for the reader to see at a glance.
Point 5: Please make a comparison of H2O2 sensing response of this work with those reported in the other literature.
Response 5: Thank you very much for your valuable suggestions. Based on your suggestion, we compared the reported H2O2 sensors in other literatures with our MoS2/RGO sensors and the results are listed in Table 1 in the revised manuscript. A brief description of Table 1 is as follows:
Table 1. Comparison of the reported H2O2 sensors and our sensors of MoS2/RGO-3.
Sensing materials | Analyte concentration | Temperature (°C) | Response (%) | Response time | Ref. |
SWCNTs | 100 ppm H2O2 | 250 | ~ 1.97 | ~ 20 s | [51] |
Pt-SWCNTs | 60.6 ppm H2O2 | 100 | ~ 50 | ~ 240 s | [52] |
CuPc-f-MWNTs CoPc-f-MWNTs VPc-f-MWNTs | 34% H2O2 (aq) vapors | Ambient temperature | (-) 24.2 (+) 3.8 (-) 4.4 | 2 s 2 s 4 s | [53] |
MoS2/RGO-3 | 50 ppm H2O2 | 25 | ~ 373.1 | ~ 9 s | This work |
The sensing performances of sensors based on MoS2/RGO-3 and other reported chemiresistive sensors for detection of H2O2 vapor can be found in Table 1. It is clearly shown that the sensor of MoS2/RGO-3 shows a relatively low operating temperature ( 25℃) and a relatively high response ( 373.1% ) for H2O2 vapor. The response time for 50 ppm H2O2 vapor is approximately 9 s. This comparison shows that our sensor of MoS2/RGO-3 has better comprehensive sensing performance for H2O2 vapor. The corresponding statements are supplemented in the revised manuscript (Lines 266-272 on Page 8).
Point 6: Please check the grammar carefully. For example, (lane 50) “With the continuous efforts to optimize the sensing performance of graphene,……, such as the structural defect[32] doped metal oxide[33] and Experiment[34] both clarified that the dangling bond defects at the edge of the graphene nanoribbon are more strongly interacting with the gas molecules…”.
Response 6: We are very sorry for the grammatical mistakes that caused problems to the manuscript. We have carefully revised the problems in the 50 lines of the manuscript. The amendments are as follows: Some important factors affecting the sensing properties of graphene were found in the process of continuous optimization of graphene sensing performances, for example, structural defects[32] and doped metal oxides[33]. Experiments[34] have shown that the defects at the edges of graphene nanoribbons interact more strongly with gas molecules, and the reduced graphene oxide (RGO) has a high defect density for fabricating gas sensors[35].(Lines 56-61 on Page 2)
Thank you once again for spending your precious time carefully reviewing our manuscripts. Your valuable suggestions will further improve and optimize our manuscripts.
Round 2
Reviewer 1 Report
Thank you for making the suggested edits and corrections for all of the reviewers's comments. I recommend publishing this article in its current form with appropriate editing for final publication.
Author Response
Reviewer 1's comments to the Author:
Thank you for making the suggested edits and corrections for all of the reviewers's comments. I recommend publishing this article in its current form with appropriate editing for final publication.
Response to Reviewer 1's Comments:
Thank you again for reviewing our manuscript, thanks to your valuable review comments, our manuscript has been greatly improved. We have carefully checked the details of the manuscript and made corresponding revisions, hoping to get approval.
Best regards!
Sincerely,

Reviewer 2 Report
The revised manuscript was suggested to publish.
Author Response
Reviewer 2's comments to the Author:
The revised manuscript was suggested to publish.
Response to Reviewer 2's Comments:
Thank you for review of the manuscript and the affirmation of the revised comments again. With your valuable suggestions, our manuscript can become more perfect.
Best regards!
Sincerely,
Reviewer 3 Report
I am satisfied that the authors have addressed all my comments but I still have one question.
In Table 1, I found that all sensors (ref.51, 52, and 53) can be used for detection of H2O2 vapors at room temperature. Please check the sensing conditions for all references again and revise the table. Moreover, please revise the following sentence,” It is clearly shown that the sensor of MoS2/RGO-3 shows a relatively low operating temperature (25℃) and a relatively high response (373.1%) for H2O2 vapor.”
Author Response
Reviewer 3's comments to the Author:
I am satisfied that the authors have addressed all my comments but I still have one question.
In Table 1, I found that all sensors (ref.51, 52, and 53) can be used for detection of H2O2 vapors at room temperature. Please check the sensing conditions for all references again and revise the table. Moreover, please revise the following sentence,” It is clearly shown that the sensor of MoS2/RGO-3 shows a relatively low operating temperature (25℃) and a relatively high response (373.1%) for H2O2 vapor.”
Response to Reviewer 3's Comments:
Thank you very much for reviewing our manuscript again. I am very sorry that due to our mistakes, there have been errors in the statistical temperature in Table 1. We have carefully checked the relevant references again and revised Table 1 accordingly, and change “It is clearly shown that the sensor of MoS2/RGO-3 shows a relatively low operating temperature (25℃) and a relatively high response (373.1%) for H2O2 vapor.” to “As can be seen clearly from the Table 1, our sensors also work at room temperature like the reported sensors, but the sensor in our work has the highest sensitivity to H2O2 vapor at room temperature, achieving a response of 373.1% to 50 ppm H2O2 vapor.” Thank you once again for your care and valuable advice, and your meticulous attitude is worth learning.
Best regards!
Sincerely,
